# Regeneration in the absence of canonical neoblasts in an early branching flatworm

Ludwik Gąsiorowski [1], Chew Chai[2], Andrei Rozanski [1], Gargi Purandare [1], Fruzsina Ficze[1], Athanasia Mizi[3], Bo Wang [2] & Jochen C. Rink [1]✉

The remarkable regenerative abilities of flatworms are closely linked to neoblasts – adult pluripotent stem cells that are the only division-competent cell type outside of the reproductive system. Although the presence of neoblast-like cells and whole-body regeneration in other animals has led to the idea that these features may represent the ancestral metazoan state, the evolutionary origin of both remains unclear. Here we show that the catenulid *Stenostomum brevipharyngium*, a member of the earliest-branching flatworm lineage, lacks conventional neoblasts despite being capable of whole-body regeneration and asexual reproduction. Using a combination of single-nuclei transcriptomics, in situ gene expression analysis, and functional experiments, we find that cell divisions are not restricted to a single cell type and are associated with multiple fully differentiated somatic tissues. Furthermore, the cohort of germline multipotency genes, which are considered canonical neoblast markers, are not expressed in dividing cells, but in the germline instead, and we experimentally show that they are neither necessary for proliferation nor regeneration. Overall, our results challenge the notion that canonical neoblasts are necessary for flatworm regeneration and open up the possibility that neoblast-like cells may have evolved convergently in different animals, independent of their regenerative capacity.

Flatworms (Platyhelminthes) are famous for their ability to regenerate missing body parts, with most ingroups capable of posterior regeneration and whole-body regeneration including the head present in at least three clades (Fig. 1a)[1]. The remarkable regenerative abilities of platyhelminths have been attributed to the presence of adult pluripotent stem cells (APSCs), the so-called neoblasts[2–4]. Originally defined based on morphology[2], molecular and functional characterizations of neoblasts in planarians[5–7], macrostomids[8,9], and neodermatans (parasitic flatworms)[10,11], revealed remarkable similarities between those groups. In all studied flatworms, neoblasts represent the only division-competent somatic cell type that resides outside of organs and gives rise to all new cells during regeneration and homeostatic tissue turnover. Neoblasts are therefore considered an important preadaptation to whole-body regeneration[2,3]. At the molecular level, neoblasts are characterized by the expression of multiple genes belonging to the germ-line multipotency program (GMP)[7,9,11–13], e.g., *piwi*, *nanos*, *argonaute*, or *vasa*, which protect against transposon-induced changes to the genome, repress somatic differentiation and are thought to organize chromatin[14–17] in order to preserve neoblast pluripotency.

Adult stem cells expressing the GMP complement and large batteries of other common genes[18] have also been observed in other well-regenerating animals, such as acoels[19–21], sponges[15,17], cnidarians[22–28], ctenophores[29], or some annelids[30]. Hence it is plausible that the last common ancestor of all Metazoa possessed neoblast-like APSCs and was capable of regeneration. Accordingly, the functional and

[1]Department of Tissue Dynamics and Regeneration, Max Planck Institute for Multidisciplinary Sciences, Göttingen, Germany. [2]Department of Bioengineering, Stanford University, Stanford, CA, USA. [3]Institute of Pathology, University Medical Centre Göttingen, Göttingen, Germany. ✉e-mail: jochen.rink@mpinat.mpg.de

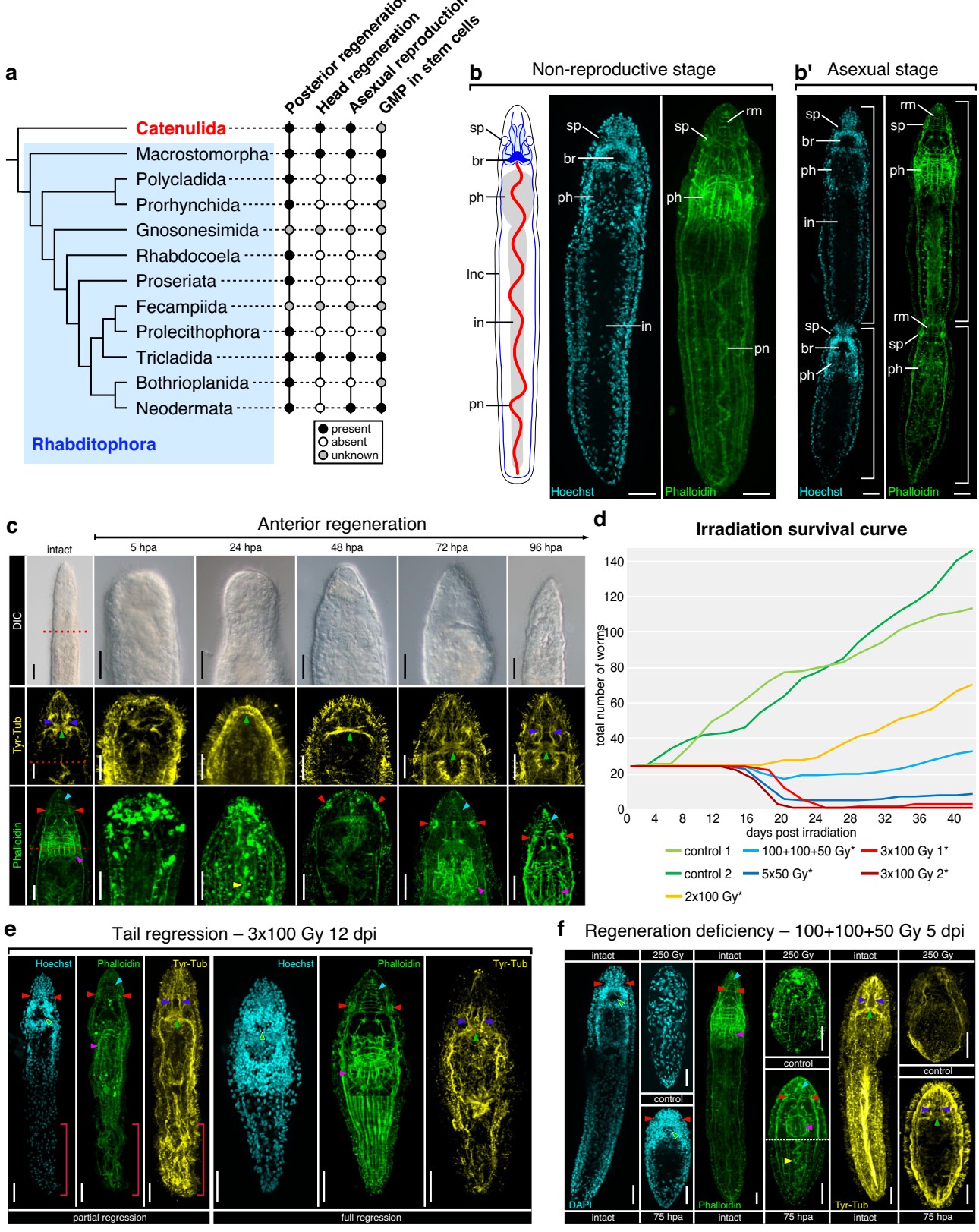

**a** Posterior regeneration, Head regeneration, Asexual reproduction, GMP in stem cells

Catenulida
Macrostomorpha
Polycladida
Prorhynchida
Gnosonesimida
Rhabdocoela
Proseriata
Fecampiida
Prolecithophora
Tricladida
Bothrioplanida
Neodermata

Rhabditophora

● present
○ absent
◐ unknown

**b** Non-reproductive stage

**b'** Asexual stage

**c** Anterior regeneration — intact, 5 hpa, 24 hpa, 48 hpa, 72 hpa, 96 hpa; DIC, Tyr-Tub, Phalloidin

**d** Irradiation survival curve

control 1, control 2, 2×100 Gy*, 100+100+50 Gy*, 5×50 Gy*, 3×100 Gy 1*, 3×100 Gy 2*

days post irradiation

total number of worms

**e** Tail regression – 3×100 Gy 12 dpi

Hoechst, Phalloidin, Tyr-Tub; partial regression, full regression

**f** Regeneration deficiency – 100+100+50 Gy 5 dpi

intact, 250 Gy, control, 75 hpa; DAPI, Phalloidin, Tyr-Tub

mechanistic similarities between flatworm neoblasts and APSCs in other lineages would reflect the conservation of the ancestral condition[17,19]. Alternatively, it is also possible that flatworm neoblasts might have evolved independently[2], with molecular and functional similarities to the APSCs of other animals resulting from the convergent co-option of the same molecular pluripotency program.

Therefore, at the current state of phylogenetic sampling, the evolutionary origins of GMP-expressing APSCs remain obscure[2,31].

A major challenge in elucidating the role of APSCs in the evolution of flatworm regeneration is the limited information on the stem cell system of Catenulida, a sister group to all remaining platyhelminths (i.e., Rhabditophora; Fig. 1a)[32,33]. Although catenulids have a crucial

**Fig. 1 | Regeneration and asexual reproduction in *Stenostomum brevipharyngium*. a** phylogentic distribution of regenerative abilities, asexual reproduction, and adult pluripotent stem cells in flatworms; Catenulida, which includes *S. brevipharyngium*, are marked in red. **b** asexual reproduction by paratomy; br, brain; in, intestine; lnc, longitudinal nerve cord; ph, pharynx; pn, protonephridium; rm, rostral muscles; sp, sensory pits; white bars indicate two zooids. **c**, anterior regeneration; red dotted lines indicate amputation planes. **d**–**f** effects of irradiation on paratomy and survival (**d**), tissue homeostasis (**e**), and regeneration (**f**).

Morphological structures (**c**, **e** and **f**) are labeled with arrowheads as follows: dark blue, rostral nerves; green, brain neuropile; red, sensory pits; cyan, rostral muscles; magenta, pharynx; yellow, protonephridium. Asterisks in (**d**) indicate significant differences from controls (at *p*-value < 0.05) as inferred with the two-sided Mann–Whitney *U* test (the exact *p*-values are provided in Supplementary Data 7). Red bars in (**e**) indicate the disorganization of posterior tissues. Scale bars on all panels represent 20 μm. Source data for panel **d** are provided as a Source Data file.

position for reconstructing ancestral flatworm conditions and are capable of full-body regeneration[1,34–37], their division-competent cells have only been studied morphologically[34,35,38,39], while their function or molecular signatures remains unknown.

Here, we provide the first comprehensive description of the stem cell system in a catenulid, *Stenostomum brevipharyngium*, to shed new light on its role in the regeneration process and, more generally, on the characteristics and evolution of the APSCs in animals. We show that catenulids lack conventional neoblasts despite being capable of whole-body regeneration and asexual reproduction, challenging the notion that canonical neoblasts are necessary for flatworm regeneration.

## Results

### *Stenostomum* tissue dynamics

To analyze the interplay between regeneration and stem cells in catenulids, we took advantage of a fortuitous laboratory culture that we identified as *S. brevipharyngium* (referred to as *Stenostomum* from here onwards), which, like many catenulid species[34,35,37,38,40], is capable of asexual reproduction by paratomy (Fig. 1b) and whole-body regeneration (Fig. 1c)[36]. Upon anterior amputation, the wound is closed by the action of circular muscles followed by the extensive remodeling of the remaining muscle fibers at the wound site by 5 h post-amputation (hpa) (Fig. 1c). Neuronal strucures, including the brain commissure and sensory organs, are regenerated between 24 and 72 hpa, while the anterior musculature (rostral muscles and pharynx) is reestablished between 72 and 96 hpa (Fig. 1c). After four days, the entire head is fully regenerated (Fig. 1c).

To assess the contribution of newly generated cells to the regeneration process, we ablated division-competent cells by X-ray irradiation. Irradiation induces double-strand breaks in DNA, which are particularly toxic to dividing cells[41]. Given the privileged division competence of flatworm neoblasts, irradiation has been used extensively for experimental neoblast ablation[6,9,42]. Neoblast ablation in macrostomids, planarians and schistosomes blocks regeneration and eventually kills the animals due to the failure of homeostatic cell turnover[9,11,42,43]. Similarly, in *Stenostomum* we observed dose-dependent lethality and an arrest of asexual reproduction in response to irradiation (Fig. 1d). However, the lethal dose was approximately four times higher than in planarians[44] and twice higher than for *Macrostomum*[9,42] and *Schistosoma*[11], thus indicating an extraordinary degree of irradiation resistance in *Stenostomum*, even by flatworm standards. Phalloidin and tyrosinated-tubulin staining of lethally irradiated worms at 12 days post-irradiation (dpi) revealed severe tissue regression indicative of disrupted cell turnover (Fig. 1e), particularly in the posterior body region. Furthermore, head regeneration was completely blocked in irradiated animals (Fig. 1f). These experiments demonstrate that adult *Stenostomum*, like their rhabditophoran sister group, rely on dividing cells for tissue maintenance, regeneration, and asexual reproduction.

### Irradiation-sensitivity of gene expression

To gain first insights into the transcriptional profile of the dividing cells in *Stenostomum*, we took advantage of the rapid elimination of dividing cells by X-ray irradiation[9,44]. RNAseq on *Stenostomum* subjected to increasing dosages of X-ray irradiation and at different time points post-irradiation (Supplementary Fig. 1a) identified 2378 irradiation-sensitive genes, 675 of which were significantly up and 1702 down-regulated at all time points (Fig. 2a, b and Supplementary Data 1). Intersecting the irradiation-sensitive *Stenostomum* genes with published sets of irradiation-sensitive genes in two rhabditophorans, *Macrostomum lignano* and *Schmidtea mediterranea* (see "Methods" for details), we identified 101 genes that are shared by all three species, thus representing the core flatworm irradiation-sensitive gene set (Supplementary Fig. 1b and Supplementary Data 2). The functional annotations of this conserved gene set and the *Stenostomum* genes most strongly downregulated at 72 h post-irradiation (hpi) both revealed enrichments for components of nucleotide metabolism, mitosis, and ribosome biogenesis (Supplementary Fig. 1c, d). Furthermore, 139 genes with a conserved function in eukaryotic ribosomal biogenesis[45] were progressively downregulated in response to irradiation (Supplementary Fig. 2a), suggesting that ribosomal biogenesis may be largely restricted to dividing cells in *Stenostomum*, as in schistosomes[46,47]. Surprisingly, our core flatworm irradiation-sensitive gene set did not include the components of the germline multipotency program (GMP), which are broadly used as flatworm neoblast markers[7,9,11,42,44]. Although the GMP homologs were present in our *Stenostomum* transcriptome, they did not show a systematic dose-dependent decrease in expression level, which was evident for example for the conserved cell division genes (Fig. 2c). In fact, *nanos* was strongly upregulated by irradiation, while one of the *piwi* paralogs (*piwiA*), two *argonaute* paralogs, two *tudor* paralogs, and *pumilio* either displayed inconsistent expression changes or were largely irradiation-insensitive (Fig. 2c). Collectively, those results provided the first indication of a divergent transcriptional signature of the dividing cells in *Stenostomum*.

### Anatomical location of dividing cells

Next, we examined the anatomical location of the dividing cells in *Stenostomum*. Whole mount visualizations of 5-Ethynyl-2′-deoxyuridine (EdU) incorporation into replicating DNA during a 1 h pulse, followed by immediate fixation, revealed specific nuclear labeling of ca. 60 cells in control animals (Fig. 2d, e), but very few or no positive cells after lethal irradiation (Fig. 2e). EdU+ cell doublets started to appear only after 24 h of incubation (Supplementary Fig. 3), indicating that our short pulse protocol only labeled cells currently in S-phase and not their division progeny. Immunocytochemistry with the M-phase cell cycle marker Ser10-phosphorylated histone H3 (H3P) resulted in similarly irradiation-sensitive cell labeling, thus confirming the specificity of both assays (Fig. 2e). Close examination of EdU incorporation revealed that the majority of dividing cells were located within two bands lateral to the intestine (Fig. 2d). EdU+ and H3P+ cell nuclei exhibited a characteristic inner cavity that could be stained with an antibody against Fibrillarin (Fig. 2f), a conserved nucleolar protein involved in ribosome biogenesis[48]. The prominence of the nucleolus in the EdU+ cells was entirely consistent with the previously noted radiation-sensitivity of ribosome biogenesis genes and was also reminiscent of the ultrastructure of the neoblasts in *Macrostomum* and *Schistosoma*[8,47,49]. In addition, we frequently observed individual EdU+ cells in the head region, the outermost cell layer (epidermis), or within the characteristic line-like arrangement of nuclei within the dorsal

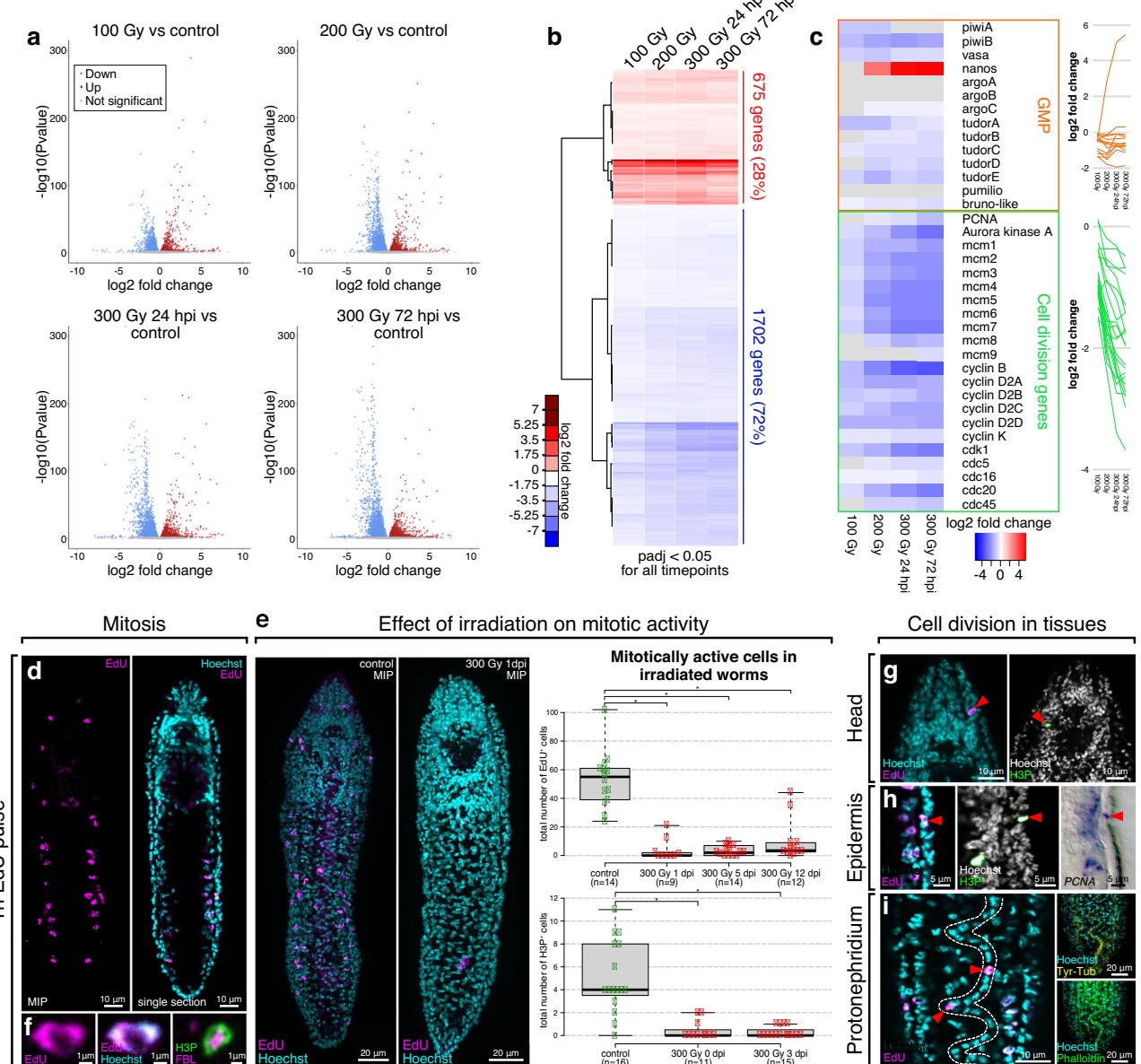

**Fig. 2 | Effects of irradiation on gene expression and cell division in *Stenostomum brevipharyngium*.** **a** Volcano plots showing statistical significance (two-sided Wald test) and magnitude of change for differentially expressed genes (DEG) in response to different irradiation doses. **b** heat map of the DEG that show statistically significant changes (adjusted *p*-value < 0.05, two-sided Wald test adjusted with the procedure of Benjamini and Hochberg) at all investigated conditions. **c** heat map and line plots showing expression level changes of germline multipotency program (GMP) components and cell-division genes in response to irradiation, gray boxes indicate insignificant changes (adjusted *p*-value > 0.05, two-sided Wald test adjusted with the procedure of Benjamini and Hochberg). **d** anatomical distribution of EdU⁺ cells. **e** effects of irradiation on mitotic activity as

inferred from EdU and H3P stainings. Center lines show the medians, box limits indicate the 25th and 75th percentiles, and whiskers extend to minimum and maximum values. Bars with asterisks indicate significant differences (at *p*-value < 0.05) as inferred with the two-sided Mann–Whitney *U* test (For EdU staining: control vs 300 Gy, *p*-value = 0.000075984; control vs 5 dpi, *p*-value = 0.0000071517; control vs 12 dpi, *p*-value = 0.00006619; for H3P staining: control vs 0 dpi, *p*-value = 0.00014; control vs 3 dpi, *p*-value = 0.00000965). Source data are provided as a Source Data file. **f** prominence of the nucleolus in division-competent cells as visualized with antibody staining against the nucleolar protein fibrillarin. **g–i** distribution of mitotically active cells (red arrowheads) in the head, epidermis, and protonephridium. MIP stands for maximum intensity projection.

protonephridium (Fig. 2g–i). H3P co-staining (Fig. 2g, h) and whole mount in situ hybridization expression patterns of the DNA replication component *PCNA* (Fig. 2h) or of multiple irradiation-sensitive genes out of our RNAseq analysis (Supplementary Fig. 2a and Supplementary Data 3) additionally confirmed the existence of dividing cells in multiple tissues of *Stenostomum*. This observation was unexpected, since in other flatworms, neoblasts as the only division-competent cell type, are excluded from differentiated tissues and organs[2,3]. Taken together with the inconsistent expression response of the GMP components to

irradiation, our results so far indicate a divergent stem cell system architecture in *Stenostomum*, when compared to other flatworms.

## Single-cell atlas of *Stenostomum* cell types

To better characterize the heterogeneity of division-competent cells in *Stenostomum*, we performed whole-body single-nuclei RNA sequencing (see Methods). After removing doublets and ambient RNA contributions, 4864 single nuclei passed the quality control, expressing a median of 1787 genes and 2674 UMI per nucleus. Self-assembling

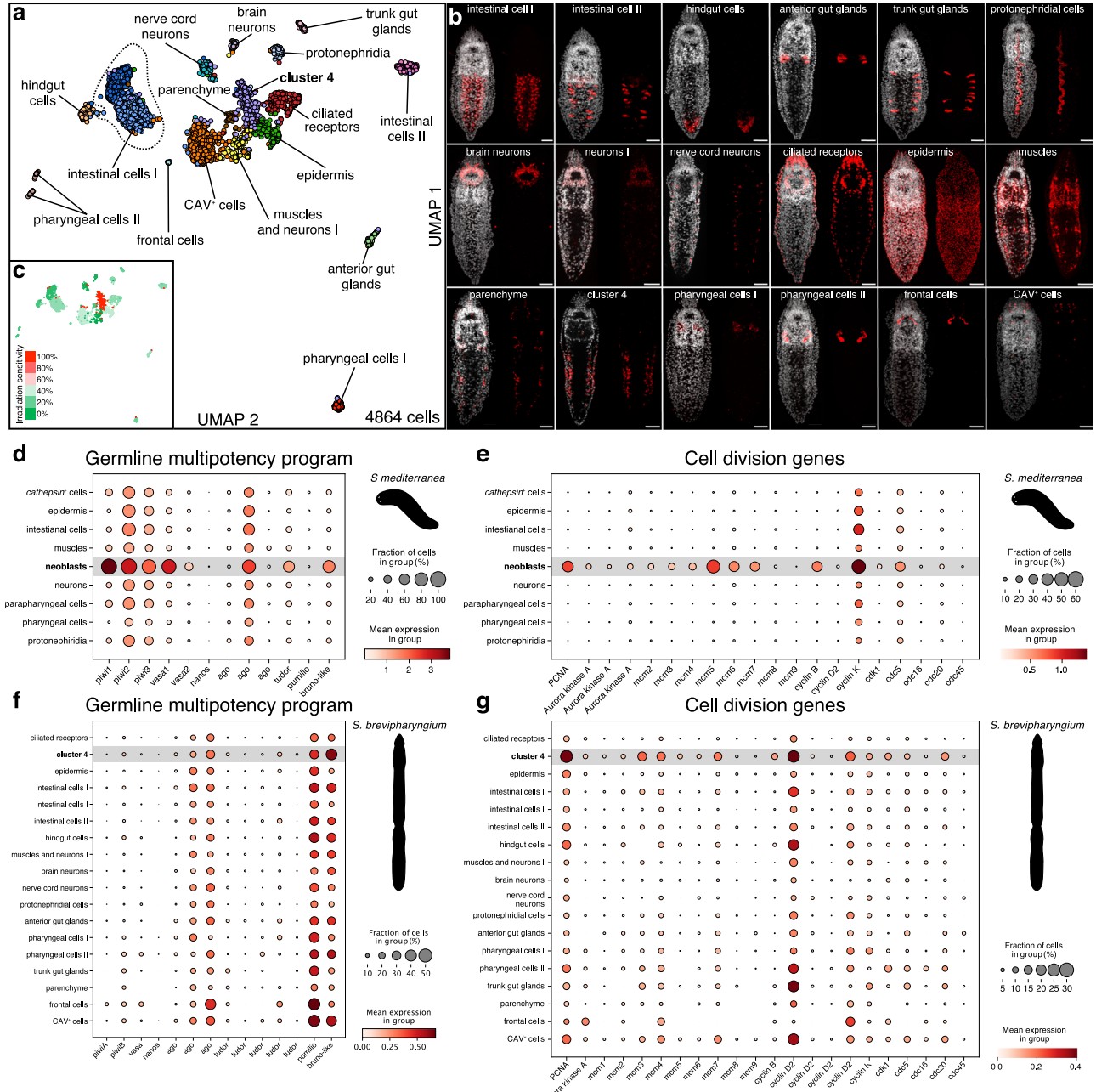

**Fig. 3 | Single-cell atlas of *Stenostomum brevipharyngium*. a** a two-dimensional uniform manifold approximation and projection (UMAP) showing cell clusters of *S. brevipharyngium*. **b** expression patterns of cluster-specific molecular markers (red), overlayed with nuclear Hoechst staining (white). Scale bars represent 20 μm. The markers used for visualizing anatomical cell type distributions are specified in Supplementary Data 3. The newly identified gene *C4M1* (go_Sbre_v1_33587_1_1) was used as a marker for cluster 4. **c** irradiation sensitivity of particular cell clusters. **d–g** dotplots showing cell type-specific expression of the components of the germline multipotency program (**d**, **f**) and cell division genes (**e**, **g**) in *Schmidtea mediterranea* (**d**, **e**) and *S. brevipharyngium* (**f**, **g**).

manifolds (SAM) analysis[50] and Leiden clustering divided them into 18 distinct clusters (Fig. 3a), which we then annotated (Supplementary Data 4 and 5) and validated through in situ hybridization chain reaction (HCR) (Fig. 3b). Collectively, our dataset represents the first single-cell atlas of a catenulid, containing intestinal, epidermal, neuronal, protonephridial, muscular, parenchymal, pharyngeal, and glandular cell types (Fig. 3a, b and Supplementary Fig. 4a and Supplementary Data 4).

To associate division-competence with specific cell clusters, we first examined the proportion of irradiation-sensitive genes amongst the 100 top marker genes for each cluster (Fig. 3c). This analysis revealed a single cluster (cluster 4) with an extreme degree of irradiation sensitivity (99%) in comparison with all other clusters (0–38%;

Fig. 3c and Supplementary Fig. 4c). The expression profiles of cells in this cluster were enriched for cell division- and ribosomal biogenesis-related genes (Supplementary Data 4), consistent with our previous bulk RNAseq analysis. The HCR expression patterns of select cluster 4-specific genes were mostly lateral to the intestine and in epidermal bands (Fig. 3b), clearly segregated into different anatomical locations consistent with the EdU incorporation and fibrillarin antibody staining patterns (Supplementary Fig. 4d and see below). In addition, multiple cell division-specific genes were broadly expressed in multiple clusters (see below).

To compare our *Stenostomum* cell atlas with an asexual *S. mediterranea* dataset[51], we used SAMap method[18]. Cluster 4 had the highest

similarity to planarian neoblasts, with 275 *Stenostomum* genes supporting this mapping (Supplementary Data 6). This gene list was again heavily dominated by ribosomal biogenesis genes (39%) and additionally contained many genes relating to cell division and DNA replication (7%), but it lacked GMP components except for a single *argonaute* paralog (go_Sbre_v1_32916_2_1, Supplementary Data 6). In addition, two transcription factors, *Sox*-like and *klf*-like, were shared between cluster 4 and planarian neoblasts as well as stem cells of other metazoa[18]. Thus, although the *Stenostomum* cluster 4 and planarian neoblasts share the gene expression signatures of dividing and metabolically active cells, they do not share canonical neoblast markers, such as GMP genes.

To follow up on these observations, we next compared the partitioning of GMP components and cell-division markers across all *Stenostomum* and *S. mediterranea* cell types. Whereas both gene sets are enriched in planarian neoblasts (Fig. 3d, e), the GMP components in the *Stenostomum* transcriptome were either broadly expressed across many cell type clusters (*argonaute*, *pumilio*, *Bruno*-like) or not detected (e.g., two *piwi* paralogs; Fig. 3f). Similarly, cell division genes including *PCNA*, *MCM* homologs and multiple cyclins, were broadly expressed across multiple *Stenostomum* cell clusters (Fig. 3g). Interestingly, our single nuclei analysis also confirmed the specific expression of ribosomal biogenesis genes in cluster 4, which is different from planarian neoblasts, but reminiscent of the stem cells of schistosomes[46] (Supplementary Fig. 4e, f). Overall, our single-nuclei analysis confirmed that the *Stenostomum* stem cells are very different from those in other flatworms: rather than a single cluster of GMP-expressing and mesenchymally located division-competent neoblasts, *Stenostomum* likely maintains division-competent cells in multiple tissues.

### Complexity of the *Stenostomum* stem cell system

To further probe the identity of the division-competent cells in *Stenostomum*, we examined the whole mount HCR expression patterns of three marker genes that were highly enriched in Cluster 4. The first marker, go_Sbre_v1_33587_1_1 (nicknamed *cluster 4 marker 1*, *C4M1*), represents a likely catenulid-specific protein, as our homology searches revealed only one clear ortholog in the published transcriptome[32] of *Stenostomum leucops* (Supplementary Fig. 5a), which was absent in other flatworms or metazoans. C4M1 contains highly repetitive stretches of negatively charged amino acids and two predicted RNA recognition motif domains (Supplementary Fig. 5a, b; see "Methods" for details). The other two markers include the ribosomal biogenesis factor *fibrillarin* (FBL) and histone *H2A* (H2A). All three markers were clearly co-expressed in the division-competent cells along the gut and in the epidermis (Fig. 4a, b and Supplementary Fig. 6a–c) that incorporated EdU during a 2 h pulse (Fig. 4a–d and Supplementary Fig. 6e, f), thus confirming both the inferred division-competence of cluster 4 and the anatomically heterogeneous distribution of its constituent cells. Therefore, we designated the cluster 4 sub-fractions as prospective "deep" and "epidermal" stem cells, respectively.

To test whether the deep and epidermal stem cells are also transcriptionally distinguishable, we examined the co-expression of *H2A* and *C4M1* with epidermal markers (*rootletin* and go_Sbre_v1_33529_1_1 – *epidermal marker 1*, *EM1*) by HCR. As expected, the deep *H2A*⁺/*C4M1*⁺ cells did not express the epidermal markers at detectable levels (Fig. 4e, f and Supplementary Fig. 6a–c). However, 2 h-pulse EdU-labeled cells in the epidermis were clearly positive for both marker sets (Fig. 4e, g and Supplementary Fig. 6g), indicating that cluster 4 marker genes can be expressed within the epidermis. Concordantly, cluster 4 could be subclustered into rootletin low/high fractions (Supplementary Fig. 4c), indicating that cluster 4 is composed of multiple division-competent cell populations. To more broadly explore the degree of overlap between division and cluster 4 marker expression in different tissues, we first examined the expression of the protonephridial marker SLC4A7 together with C4M1 and 2 h EdU pulse incorporation.

Interestingly, the EdU⁺/*SLC4A7*⁺ cells in the protonephridium did not express *C4M1* at levels detectable by HCR (Fig. 4h, i). Moreover, intestinal cells, marked by expression of go_Sbre_v1_44801_2_1 (designated *gut marker 1*, *GM1*), that showed a low degree of irradiation-sensitivity in our single nuclei analysis (Supplementary Fig. 4c), neither incorporated a 2 h EdU pulse nor expressed the cluster 4 markers *H2A* or *C4M1*. In fact, the expression of *GM1* and *H2A*/*C4M1* was mutually exclusive (Fig. 4j and Supplementary Fig. 6b). Therefore, dividing cells in different organs may exhibit different transcriptional signatures and not all organs contain dividing cells. These results reveal an unexpectedly complex stem cell system in *Stenostomum*, comprising the presumptive deep stem cells that express catenulid-specific markers, the presumptive epidermal stem cells that co-express stem cell and differentiated cell marker genes, division-competent cells in other tissues that lack stem cell markers (e.g., in the protonephridium), and organs that are likely devoid of division-competent cells (e.g., intestine).

### GMP expression in gonadal cells

Besides the broad distribution of division-competence amongst *Stenostomum* cells, the above-noted lack of consistent GMP complement expression by any of the dividing cell types represents a further stark difference to other flatworms. To corroborate these findings and to test whether other cell types might express the GMP components, we designed HCR probes for the canonical GMP components: *piwi* (two paralogs present in *Stenostomum*), *vasa*, and *nanos* (both represented by single paralogs). One of the *piwi* paralogs (*piwiA*) and *vasa* were robustly co-expressed in small and sparse clusters of epidermal cells arranged in a zigzag pattern along the dorsal midline (Fig. 4k and Supplementary Fig. 6c, d), which did not express *C4M1* and were not labelled by 2 h EdU pulses (Fig. 4l and Supplementary Fig. 6c). Therefore, it is likely that the relatively rare dorsal *piwiA*⁺/*vasa*⁺ cells were not captured in our single nuclei atlas. *nanos* expression was also specific to *piwiA*⁺ cells, but only detectable in a subset of the cells and, interestingly, only in specimens in the process of asexual reproduction (Supplementary Fig. 6h). The second *piwi* paralog (*piwiB*) was not co-expressed with the other components of the GMP complement and, consistent with its irradiation-sensitivity (Fig. 2c), was expressed in a small subset of division-competent cells in the lateral epidermis (Fig. 4m).

Some animals, including the annelid *Platynereis dumerilii*, transiently activate GMP expression at wound sites to facilitate cell dedifferentiation during regeneration[52]. To test whether the *Stenostomum* GMP complement might be similarly dynamically deployed in the context of regeneration, we examined the expression of the GMP and of stem cell markers in head-regenerating animals at 1, 5, 12, and 24 hpa (Fig. 4n). While *C4M1*⁺ and *FBL*⁺ cells could be detected at the wound site from 5 hpa onwards, the GMP components did not show any changes to their steady-state expression patterns, indicating that the GMP complement is not transiently expressed in regeneration-associated cell types in *Stenostomum*.

To follow up on our previous observation of increased *nanos* expression upon irradiation (Fig. 2c), we examined the HCR expression patterns of GMP complement genes in lethally irradiated animals (300 Gy 72 hpi). As shown in Fig. 4n, the *piwiA*⁺/*vasa*⁺ cell clusters were morphologically unaffected by irradiation, in stark contrast to the irradiation-induced ablation of *C4M1* expressing cells (Supplementary Fig. 6i) or the GMP⁺ neoblast populations of other flatworms[7,11,42]. Moreover, the *nanos* signal in irradiated dorsal piwiA⁺ cell clusters increased dramatically and, interestingly, these cells simultaneously became positive for the stem cell marker *C4M1* (Fig. 4o). In light of previous findings demonstrating the stress-induced differentiation of transitory gonads from epidermal cells in catenulids[53], we conclude that the dorsal GMP⁺ cell clusters likely represent the latent gonadal anlagen that initiate their development upon irradiation stress. Overall, our results suggest that the GMP complement expression is restricted to the germ line in *Stenostomum*.

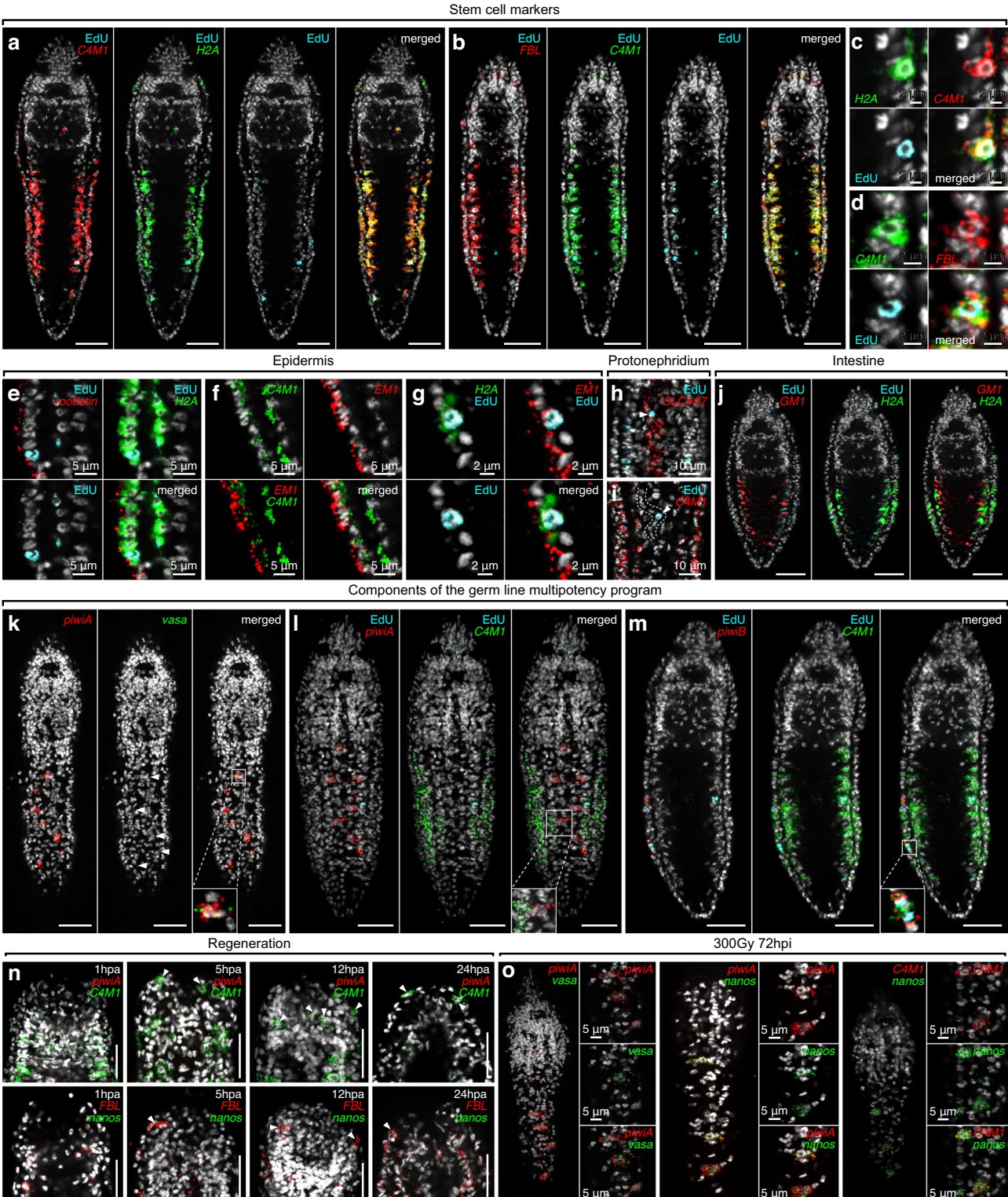

**Fig. 4 | Gene expression and division competence in stem cells, somatic tissues, and germline of *Stenostomum brevipharyngium*. a–d** expression of the stem cell markers derived from single-cell experiments combined with 2 h EdU incorporation. **e–g** co-expression of epidermal and stem cell markers in dividing cells of the epidermis. **h, i** EdU⁺ cells (arrowheads) in the protonephridium express the protonephridial marker (**h**), but not the stem cell marker (**i**). **j** mutually exclusive expression of gut and stem cell markers. **k–m** expression of the germline multipotency program (GMP) components and stem cell markers, the presumptive gonadal anlagen on the dorsal side of the animal express GMP components, *piwiA* and *vasa* (arrowheads, **k**), but not the stem cell marker (**l**). **n** cells expressing stem cell markers (arrowheads) but not GMP components are present in the blastema of head-regenerating worms. **o** elevated expression of GMP components and stem cell markers in gonadal anlagen of lethally irradiated worms. Cell nuclei are counterstained with Hoechst (white). Scale bars on all panels represent 20 µm, if not indicated otherwise.

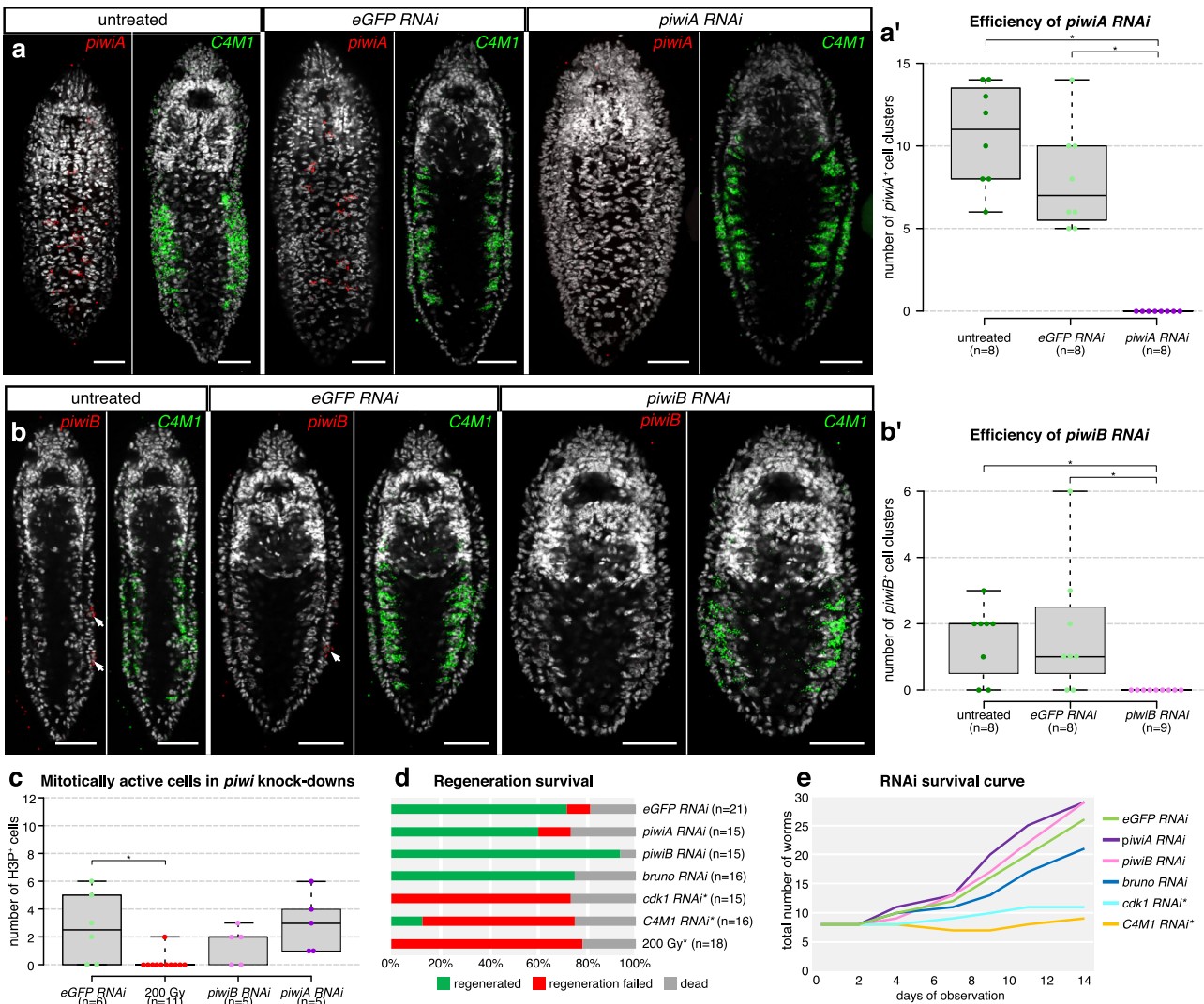

**Fig. 5 | Functional requirements of GMP and stem cell marker genes in asexual reproduction and regeneration of *Stenostomum brevipharyngium*.**
**a**, **b** validation of the efficiency of the dsRNA-mediated knockdown of *piwiA* (**a**) and *piwiB* (**b**), cell nuclei are counterstained with Hoechst (white), scale bars represent 20 μm. **c**–**e** effects of gene knockdowns on mitotic activity (**c**), anterior regeneration (**d**), survival and asexual reproduction (**e**). Bars with asterisks indicate significant differences (at *p*-value < 0.05) as inferred with the two-sided Mann–Whitney *U* test (*piwiA RNAi* vs eGFP, *p*-value = 0.00094; *piwiA RNAi* vs

untreated *p*-value = 0.00094; *piwiB RNAi* vs eGFP, *p*-value = 0.01078; *piwiB RNAi* vs untreated *p*-value = 0.01078). Asterisks indicate a significant difference from the *eGFP RNAi* control group (at *p*-value < 0.05) as inferred with the two-sided Fisher's exact test (**d**) and two-sided Mann–Whitney *U* test (**e**); the exact *p*-values for both comparisons are provided in Supplementary Data 10, 11. In the boxplots, center lines represent the median, box limits indicate the 25th and 75th percentiles, and whiskers extend to minimum and maximum values. Source data for (**a**–**e**) are provided as a Source Data file.

## Gene function analysis

Finally, we thought to complement our marker analysis with functional tests of GMP component and stem cell gene requirements during regeneration, paratomy and homeostasis. We established RNA interference by double stranded RNA soaking (dsRNAi), inspired by existing protocols in *Macrostomum*[9] (see Methods). The addition of dsRNA targeting *piwiA*, *piwiB* or *C4M1* to the culture medium quantitatively depleted the HCR signal of the targeted genes (Fig. 5a, b and Supplementary Fig. 6j) and reduced the counts of positive cells (Fig. 5a, b), thus demonstrating that *Stenostomum* is susceptible to RNAi by soaking. Strikingly, the expression of the stem cell marker *C4M1* was unaffected by *piwiA* or *piwiB* RNAi (Fig. 5a, b), showing that neither *piwiA* nor *piwiB* are acutely required for the maintenance of the *C4M1*+ division-competent cells, even though *piwiB* is expressed in subpopulations of those cells. In addition, the numbers of H3P+ cells, as a proxy for mitotic activity, were unaffected by *piwiA* or *piwiB* RNAi (Fig. 5c), very different from the specific requirement for specific *piwi*

paralogs in maintaining cell proliferation in planarians[7]. To broaden our gene function analysis, we included the additional GMP component *bruno-like*, and the conserved cell cycle regulator *cdk1* into our RNAi screen and assessed head regeneration and paratomy as a readout for stem cell function at the organismal level. As expected, *cdk1(RNAi)* was strongly inhibitory, mirroring the effects of lethal irradiation (Fig. 5d, e). The very similar phenotype of *C4M1(RNAi)* revealed its requirement for stem cell function and, in addition, that at least one of the *C4M1*+ cell types is likely to be required for regeneration. In stark contrast, none of the tested GMP components had any effect on regeneration or paratomy.

Overall, our results demonstrate that the GMP components are not required for asexual reproduction, tissue homeostasis, or regeneration in *Stenostomum*. This implies that in *Stenostomum*, in contrast to other flatworms, the GMP is restricted only to the putative germline precursors, while remaining spatially and functionally decoupled from the somatic stem cell system.

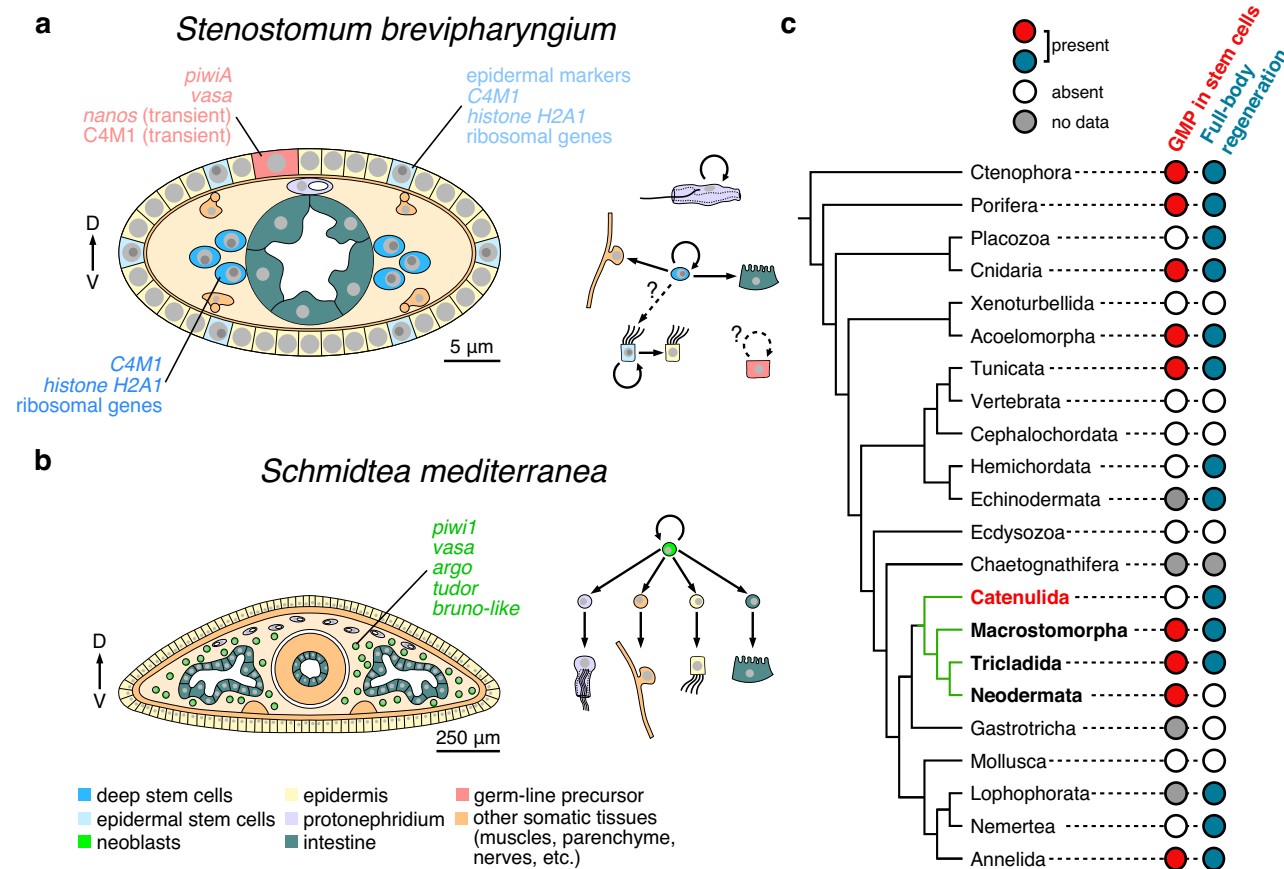

**Fig. 6 | Divergence of flatworm stem cell systems in the context of metazoan stem cells and regeneration. a, b** Idealized cross-sections through the trunks of *Stenostomum brevipharyngium* (**a**) and *Schmidtea mediterranea* (**b**) compare their respective stem cell systems. Arrows indicate the dorso-ventral axis of the animals. *S. brevipharyngium* relies on a complex stem cell system with multiple division-competent cell types, while in *S. mediterranea,* all somatic cell types are derived from neoblasts (green) as only division-competent cell type that is molecularly defined by germline multipotency program (GMP) expression. **c** distribution of the GMP-positive stem cells and regenerative abilities across metazoan phylogeny. The flatworm clade is indicated in green.

## Discussion

Our results show that the *Stenostomum* stem cells (Fig. 6a) are strikingly different from the neoblasts found in other flatworms (Fig. 6b). The widely accepted operational definition of neoblasts is that they are APSCs, and the only division-competent cell type outside of the reproductive system that gives rise to all other cell types during homeostasis and regeneration. Molecularly, neoblasts are defined by constitutive expression of the components of the GMP (e.g., *piwi*, *nanos*, *argonaute*) and anatomically they are restricted to the mesenchymal space surrounding all internal organs[11,54,55]. The stem cell system that we found in *Stenostomum* differs in virtually every aspect of this definition: instead of a single division-competent cell type, there are multiple division-competent cell populations, some located within differentiated tissues and organs. Furthermore, none of them expresses the full GMP complement and none of the tested GMP components are necessary for cell division or regeneration. Instead, the full GMP complement is only expressed in the presumptive germline. Although we cannot assess the potency of dividing cell types in *Stenostomum* due to technical limitations, it is plausible that the multiple division-competent cells represent multipotent or lineage-restricted stem cells and that *Stenostomum* lacks APSCs. Another possibility is that one of the division-competent cell types (e.g., the deep stem cells within cluster 4) retains neoblast-like pluripotency and gives rise to the other (stem) cell types. However, the gene expression overlap between cluster 4 and planarian neoblasts is largely limited to generic cell division and metabolism genes, implying that pluripotency would have to be specified independently of GMP expression in catenulids. Either

way, we can conclude that *Stenostomum* does not possess a canonical neoblast-based stem cell system, despite being capable of asexual reproduction and whole-body regeneration[1,36,37,40].

Given the phylogenetic position of catenulids as a sister group to all other flatworms[32,33], our results raise important evolutionary questions regarding neoblasts, pluripotency, GMP, and regeneration. The broad phylogenetic distribution of neoblast-like cells and their presence in several early-branching animal lineages (Fig. 6c) are often taken as evidence for the presence of GMP-expressing APSCs in the last metazoan common ancestor[15,17,19,27]. Accordingly, the presence of neoblasts in all flatworms except catenulids could reflect a specific loss in catenulids, accompanied by the acquisition of divisional competence by multiple somatic cell types and the restriction of GMP to the germ line. Alternatively, the catenulid stem cell system could resemble the plesiomorphic platyhelminth conditions, while GMP co-option to APSC and loss of tissue-associated stem cells in other flatworms represent the secondary derived state. Although such a scenario may seem controversial, on closer inspection, the presence of GMP⁺ stem cells has only been reported from a handful of animal taxa, scattered across phylogeny and interspersed with clades that either do not have neoblast-like cells or in which molecular characterizations of the stem cell systems have not yet been attempted (Fig. 6c). Importantly, the possibility that neoblasts are a derived state within flatworms would therefore imply that the broad similarities between flatworm neoblasts and APSCs in other animals may be the result of convergent evolution rather than evolutionary conservation.

Although the current phylogenetic sampling of animal APSCs is too sparse to exclude either possibility[31], our results show that the evolution of stem cell systems in flatworms and other lineages is more complex than previously thought. In particular, our results highlight significant evolutionary plasticity in the distribution of mitotic competence, pluripotency maintenance, and GMP deployment between soma and germline. Future detailed mechanistic studies are warranted for clades that either occupy crucial phylogenetic positions (e.g., chaetognathiferans, gastrotrichs) or show excellent regenerative capacities (e.g., hemichordates, lophophorates, nemerteans). These studies will help clarify how many times neoblast-like cells have evolved and their evolutionary relationship to the hypothetical APSCs of the common metazoan ancestor. Understanding the link between GMP expression and pluripotency maintenance in different lineages will shed light on the role of APSCs in the evolution of whole-body regeneration.

## Methods

### Animal husbandry and morphological investigation

The founding animals of our *Stenostomum* colony were ordered in 2010 from Connecticut Valley Biological Supply as *Stenostomum* sp. and since then maintained in the laboratory. The cultures were maintained in Chalkley's Medium (CM) at 20 °C in the dark and fed *ad libitum* with the unicellular eukaryote *Chilomonas* sp. as previously described[36]. Amputations also followed the same procedure as previously described[36].

For morphological investigations of intact and manipulated worms, we also followed an established protocol[36]. We used mouse anti-tyrosinated tubulin primary antibodies, Sigma T9028 (at 1:500 dilution), and goat anti-mouse secondary antibodies conjugated with Alexafluor488, Thermo Fisher A-11001 (at 1:250). To stain cell nuclei, we used DAPI (at 1:1000) or Hoechst 33342 (at 1:5000), and for actin filament staining we used Phalloidin, conjugated with Alexafluor555, Thermo Fisher A34055 (at 10 μ/ml).

### Irradiation assays

All irradiation experiments were done using a CellRad Precision benchtop X-ray irradiator. The worms were placed in a small Petri dish without a lid, irradiated with the desired dose of irradiation, and then washed with fresh CM. For multi-dose irradiation paradigms, the worms were allowed to recover for 24 h between successive irradiation treatments. For survival assays, single worms were placed in wells of a 24-well plate in fresh CM with *Chilomonas* sp. added *ad libitum*. The numbers of worms in each well (i.e., originating from a single experimental individual) were scored manually by observations with a dissecting microscope at intervals of 2 to 3 days over 6 weeks (see Supplementary Data 7 for details). To statistically test for differences between treatments, the slope of the survival curves of individual worms was calculated ($n = 24$) in each treatment and compared to the slopes of the survival curves of control worms using the two-sided Mann–Whitney $U$ test (Supplementary Data 7).

### RNA extraction and transcriptome sequencing

For RNA extraction ca. 60 worms were pooled for each replicate and preserved in RNA-later. RNA extraction was performed using the NucleoSpin RNA XS kit for RNA purification, Macherey–Nagel, 740902.50, according to the manufacturer's recommendation with the following modifications: before extraction, RNAase free PBS was added to each sample to dilute the RNA-Later reagent, and the samples were centrifuge at $4500 \times G$ for 10 min at 4 °C in a benchtop centrifuge to pelletize the worm tissues. Then the supernatant was removed and replaced with the kit's lysis buffer. To enhance the lysis, samples were snap-frozen in liquid nitrogen and then thawed in a water bath at room temperature. After this step, the samples were processed following the standard protocol for RNA isolation from tissues. The extracted RNA

was sent for sequencing on an Illumina NovaSeq 6000 at the Dresden Concept Genome Center to a depth of 40 million paired-end reads with 100 bp length.

### Differential gene expression analysis

Raw read quality control and adapters trimming were performed by FastQC (0.11.9)[56] and trimgalore (0.6.6)[57], respectively, both with standard parameters. The raw reads were mapped to the published assembled reference transcriptome of *S. brevipharyngium*[36], available at Zenodo (doi: 10.5281/zenodo.8239273). To align, estimate abundance, and run differential expression analysis we have used the Trinity suite (v 2.14.0)[58]. For the alignment and abundance estimation step the parameters used were '--est_method RSEM', '--aln_method bowtie2'. The differential expression analysis was performed using DESeq2 (v Bioconductor 3.15)[59]. Based on these results we established two sets of irradiation-sensitive genes: a stringent set (1703 transcripts)—that shows a negative value of log2fold change and padj value < 0.05 at all conditions (100 Gy, 200 Gy, 300 Gy 24 hpi and 300 Gy 72 hpi), and a relaxed set (2813 transcripts)—that shows a negative value of log2fold change and padj value < 0.05 at two last timepoints (300 Gy 24 hpi and 300 Gy 72 hpi). For the following analyzes, we used a relaxed set, to avoid a mistaken deletion of important genes from our dataset.

For the comparison of the irradiation-sensitive genes among flatworms, we used published datasets for *M. lignano*[9] and *S. mediterranea*[60]. For *M. lignano* we used the "neoblast" dataset and for *S. mediterranea* the genes that were enriched in the irradiation-sensitive cell fraction at log2fold ≥ 2. The first step in performing this search was to translate the transcriptome assemblies using Transdecoder (v 5.5.0)[61]. At the prediction phase we kept only the best translation by using the parameter '--single_best_only'. After the translation, we used orthofinder (v 2.5.4)[62] with standard parameters to search for orthologs. The translated transcriptome assemblies were annotated using eggnog-mapper (v 2.1.2)[63] with the diamond as a search strategy. For each of the *Stenostomum* irradiation-sensitive genes from the relaxed set, we checked whether there is at least one irradiation-sensitive ortholog in *Macrostomum* or *Schmidtea*. Genes that were shared between all three species were flagged as core flatworm irradiation-sensitive genes and their putative function was determined using the KEGG database[64]. For the analysis of the expression of genes involved in cell division, germline multipotency program, and ribosomal biogenesis, we identified *Stenostomum* orthologs of the conserved metazoan genes using reciprocal BLAST search. The IDs of all analyzed genes are provided in Supplementary Data 8.

### EdU and antibody staining against H3P and FBL

For EdU incorporation and staining, we used the Click-iT™ EdU Cell Proliferation Kit for Imaging, with Alexafluor647 detection reagent (Thermo Fischer, C10340), and followed the manufacturer recommendations with the following modifications: For EdU incorporation, we soaked the worms for 1 h (if not stated otherwise) in 400 μM EdU in CM with an addition of 3% DMSO. After incubation, the worms were washed several times in fresh CM, anesthetized with 1.44% $MgCl_2$ in CM (mass:volume), and fixed with 4% formaldehyde in PBS + 0.1% Tween-20 for 1 h. Incubations with the Click-iT® reaction cocktail were extended to 1 h at room temperature.

Antibody staining against phosphorylated histone H3 and fibrillarin followed an established protocol[36], but with the following modifications: After fixation, the animals were dehydrated by overnight incubation at −20 °C in 100% methanol. Before antibody staining, the worms were rehydrated with increasing concentrations of PBS + 0.1% Tween-20 and then subjected to proteinase K (NEB, P8107S) treatment at 10 μg/ml concentration for 2 min at room temperature. Following the protK treatment, we postfixed the worms in 4% formaldehyde in PBS + 0.1% Tween-20 for 15 min and proceeded to the standard antibody staining procedure. We used mouse anti-Fibrillarin primary

antibodies, Thermo Fisher MA316771 (at 1:100 dilution), rabbit anti-Histone H3 phospho S10 + T11, abcam ab32107 (at 1:1000), goat anti-mouse secondary antibodies conjugated with Alexafluor488, Thermo Fisher A-21422 (at 1:250), and goat anti-rabbit secondary antibodies, conjugated with Alexafluor647, Thermo Fisher A-21244 (at 1:250).

## Single-nuclei transcriptomics

For the extraction of single nuclei, we pooled together ca. 250 worms and applied the following protocol. Worms were lysed in 150 µL of lysis buffer (Supplementary Data 9 for buffer recipe), then passed several times through a hypodermal needle (0.4 mm) to break down the tissues, and incubated in lysis buffer on ice for ca. 15 min. The sample was then filtered through a 10 µm CellTrics filter to remove undissolved chunks of tissue and aliquoted into 10 µl. Each aliquot was washed with 10 µl of wash buffer (Supplementary Data 9), and centrifuged for 5 min at $800 \times G$ at 4 °C in a benchtop centrifuge. The supernatant was removed, and samples were re-suspended in 20 µl of wash buffer and pooled together. We assessed the quality of the preparation by staining 20 µl of the sample with propidium iodide (Sigma-Aldrich, P4170), inspecting the shape of the extracted nuclei under the microscope, and counting them in the Neubauer chamber (the estimated concentration of nuclei in the sample was ca. 320 nuc/µl). The sample was then split into two technical replicates and the single-nuclei transcriptome libraries were generated on the Chromium 10× Genomics platform, following the manufacturer's instructions. The libraries were sequenced on an Illumina NextSeq 550 to a depth of 400 million paired-end reads with 150 bp length.

## Single nuclei data analysis

Cell and molecular barcodes were tagged to sequenced reads using UMI-tools[65]. Primer and polyA sequences were trimmed using cutadapt[66]. Reads were aligned to the published transcriptome of *S. brevipharyngium*[36] using bowtie2 --sensitive parameter[67]. Using a knee-curve analysis, we removed empty droplets. Ambient RNA contamination was then removed using SoupX[68] with default parameters. UMI counts were used to perform downstream analysis. Cells with fewer than 1000 genes detected were filtered out, resulting in a final count of 4864 cells. On average, we detected 1787 genes and 2674 UMI for each cell. To account for differences in sequencing coverage, we normalized raw read counts such that each cell has a total count equal to that of the median library size for all cells. The resulting counts were then added with a pseudo count of 1 and log-transformed. 2D embedding was performed using the SAM algorithm[50] with rms as weight_mode parameter. There was no difference in the complexity of the obtained atlases between the two technical replicates, so we merged them for the following analyzes.

We applied random forest classifiers on each Leiden cluster to find markers enriched in each cell cluster and manually chose those that showed cluster-specific expression in the total dataset (Supplementary Data 4) and subclusters of cluster 7 (Supplementary Data 5). Log$_2$ fold change for those manually selected markers (Supplementary Data 4 and 5) was calculated by dividing the mean of a particular gene expression count for each cluster against the mean of all other clusters. *P*-values were derived from a one-sided Mann–Whitney *U* test to compare the expression counts of a particular gene between each cluster against all other clusters. By comparing the obtained markers with the literature, we provided provisional manual annotations for the clusters, which were later confirmed with HCR in situ. Most of the clusters were identified as unique cell types, however, there was no clear distinction in gene expression between clusters 0 and 1 (identified as intestinal cells), while cluster 7 was composed of 3 distinct subclusters that represent muscles and two different neuronal cell types (Supplementary Data 5). All annotations are provided in Supplementary Data 4.

To check for the irradiation-sensitive genes in our single-cell sequencing data, we first find the top 100 marker genes for each cell type using a random forest classifier, which we then overlap with the relaxed set of irradiation-sensitive genes downregulated in bulk-RNA sequencing.

For the planarian single-cell dataset, we took the published raw read counts from[51], and reprocessed the data as described above, except we filtered out cells with fewer than 500 genes. We then annotated cells with only major cell type annotations.

For the comparative analysis of the cell type-specific expression of genes involved in cell division, germline multipotency program, and ribosomal biogenesis we used previously identified *Stenostomum* orthologs and identified orthologous genes in the v6 reference transcriptome of *Schmidtea mediterranea*[69] with a reciprocal BLAST search, the IDs of all analyzed genes are provided in Supplementary Data 8.

## In situ RNA hybridization

Both colorimetric and chain reaction RNA in situ hybridization[70] followed the same protocol as in ref. 36. The HCR probe sets were designed using the Özpolat Lab's HCR in situ probe generator[71]. Plasmids containing cloned coding sequences and pools of HCR probes are available upon request from the authors.

## dsRNA interference

First, we generated a template for dsRNA synthesis by amplifying targeted coding sequences from cDNA using gene-specific primers with T7 overhangs and Q5 High-Fidelity DNA Polymerase (NEB, M0491S). Two rounds of PCR reactions (each with 35 cycles) were required to reach a satisfactory concentration of the targeted product (>100 ng/µL). The specificity of the PCR was tested by DNA gel electrophoresis and confirmed with Sanger sequencing. The purified PCR products were used as templates for in vitro RNA transcription performed with T7 RNA polymerase (Thermo Fisher, EP0112). The synthesized RNA was annealed by heating it to 75 °C for 3 min and then cooling down to room temperature. The concentration of the dsRNA was assessed with NanoDrop ND-1000 and its integrity was checked with gel electrophoresis.

The dsRNA was diluted in 600 µL of Chalkley's Medium (CM) with food (*Chilomonas* sp. solution) to a desired concentration of 50 ng/µL and distributed in 24 well plates. Ca. 30 worms were put in each well. 400 µL of the dsRNA solution in each well werereplaced daily for 14 days, with additional food added when *Chilomonas* were found absent in the wells. The integrity of the freshly added dsRNA was regularly checked with gel electrophoresis. Each RNA interference experiment was repeated in two technical replicates.

Following exposure to dsRNA, the worms were subjected to survival analysis similar to the previously described irradiation assays, but for a 2-week duration only (see Supplementary Data 10 for details), regeneration assays (worms cut through the pharynx and observed for 4 days), antibody staining against H3P (same procedure as described above), and in situ HCR, same procedure as in ref. 36). To statistically test for differences in asexual reproduction between treatments, we calculated the slopes of the survival curves for single experimental individuals ($n = 8$) in each treatment and compared them to the slopes of the survival curves of the *eGFP RNAi* control worms using the two-sided Mann–Whitney *U* test (Supplementary Data 9). The statistical significance of the differences in the regeneration success rate between experimental (RNAi, irradiation) and control animals was assessed using two-sided Fisher's exact test (Supplementary Data 10).

## Microscopy

Worms were imaged on an Olympus IX83 microscope with a spinning disc Yokogawa CSUW1-T2S scan head (for fluorescently labeled samples) or Zeiss Axiophot upright microscope (for life imaging and colorimetric in situ hybridization). The obtained confocal *Z*-stacks were processed for contrast and brightness and analyzed in Fiji[72].

All micrographs show representative images from at least 3 independently stained individuals.

## Analysis of the protein C4M1

The *C4M1* transcript was translated into a protein sequence using Geneious Prime (v2023.0.3). The protein sequence was then used as a query for TBLASTN searches against transcriptomes of various rhabditoforans using the Planmine database[69] and against published transcriptome of *S. leucops* (NCBI BioProject: PRJNA276469). The protein sequence was subjected to the InterPro (v98.0)[73] search for conserved domains. We also predicted the domain structure of the protein using an implementation of AlphaFold in ColabFold[74]. The obtained protein structure was used as a query in Foldseek Search[75] to identify putative functions of the conserved domains.

## Box plots

All boxplots were generated with BoxPlotR[76], with center lines showing the medians, box limits indicating the 25th and 75th percentiles as determined by the R software, and whiskers extending to minimum and maximum values. All the raw data used to generate displayed boxplots are provided in the Source Data file.

## Reporting summary

Further information on research design is available in the Nature Portfolio Reporting Summary linked to this article.

## Data availability

The raw RNAseq reads on which the *S. brevipharyngium* reference transcriptome is based have been deposited at the NCBI Sequence Reads Archive (BioProject ID PRJNA1004231). The assembled reference transcriptome is available at the Zenodo.org data repository [https://doi.org/10.5281/zenodo.8239273]. The raw RNAseq reads of irradiated worms have been deposited at the NCBI Sequence Reads Archive as BioProject PRJNA1149834. The raw single-cell transcriptomic data has been deposited at the NCBI under accession number PRJNA1156255 with the accession numbers SRR31280466 and SRR31280465, while the count tables as well as the final h5ad have been deposited at the Zenodo.org [https://doi.org/10.5281/zenodo.14184102]. Due to the large volume of the microscopic data used in this manuscript, the raw microscopy images will be stored on the File Server of the Max Planck Institute for Multidisciplinary Sciences and made available without restrictions upon request from the corresponding author. Source data are provided with this paper.

## Code availability

There is no custom code or mathematical algorithm developed for this study.

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

## Acknowledgements

All current and past members of Rink Laboratory are acknowledged for their scientific and technical support and especially Miquel Vila-Farré for his comments on the early versions of the manuscript and figures. We would like to thank Bernhard Egger and Jordi Solana for the valuable discussions on the evolution and function of the flatworm stem cell system. We also would like to acknowledge Animan Tripathi for her involvement in the optimization of the RNAi protocol and Torben Ruhwedel (Electron Microscopy facility of Max Planck Institute for Multidisciplinary Sciences) for the preparation of semi-thin *Stenostomum* sections. C.C. and B.W. are supported by an NIH grant 1R35GM138061. This research was supported by the Alexander von Humboldt Foundation (The Humboldt Research Fellowship for Postdoctoral Researchers to L.G.) and Max Planck Society (funding to J.C.R.).

## Author contributions

The experiments were conceived and designed by L.G. and J.C.R. Data acquisition was performed by G.P. (colorimetric in situ hybridization), F.F. and A.M. (preparing single-nuclei libraries), and L.G. (remaining experiments). A.R. analyzed irradiation transcriptomic data, and C.C. performed single-nuclei transcriptomic analyzes. Figures were prepared by L.G. and C.C. The manuscript was written by L.G. and J.C.R. with input from C.C. and B.W.

## Funding

## Competing interests

The authors declare no competing interests.
