## [Peer Review file · Nature Communications]

Regeneration in the absence of canonical neoblasts in an early branching flatworm

Corresponding Author: Professor Jochen Rink

Version 0:

Reviewer comments:

Reviewer #1

(Remarks to the Author)

The paper by Gasiorowski and collaborators present a challenge to the paradigm that Platyhelminthes regenerate body parts by mobilizing stem-cells (neoblasts). A thorough analysis of the regeneration mechanisms in the catenulid *Stenostomum* challenge previous assumptions on the presence/use of the neoblasts in the process. In this context, the findings offer important hints into the developmental and evolutionary roles of stem cells. The experimental strategy is well conceived, the findings well described, and the writing clear and focused.

Here I suggest some issues that (to me) need some clarification; some of them minor corrections. I find a bit complicated to describe them in the absence of line numbers in the draft. They are listed in the order in which they appear in the text:

1- In the introduction, the authors mentioned that neoblasts “revealed a remarkable degree of conservation between those groups”. I guess the author refer to “similarities”, instead of “Conservation”. Otherwise, the sentence turns to be a bit vague. Conservation of what? (morphology? gene patterns? both??)

2- I find that the irradiation experiments, common in other flatworms, need a bit of explanation. For instance, how do we know that irradiation (mostly at high doses) is not randomly killing (or altering) all types of mature cells?

3- Following my previous question, why (in the section Irradiation-sensitivity of gene expression) should we assume that “ribosomal biogenesis may be largely restricted to dividing cells in *Stenostomum*”? Maybe irradiation has specific transcriptional effects on all cells of each individual. In fact, the disrupted morphology of the irradiated specimens’ points to the possibility that many cells are affected/damaged. The conclusion that the transcriptional data is the “first indication of divergent transcriptional signature of dividing cells” is not fully guaranteed (though, still, reasonable).

4- The fact that the irradiation-sensitive genes map to different regions of the body is illuminating and serves the purpose that they are distributed in different tissues. However, it would have been a nice complement to show a few sections (semithin), with a more detailed analysis of the cells involved and their relative positions in the tissues. The double in situ with specific tissue markers offers some hints (though, of course, miss the resolution of a good histological section). We are left with the idea that the sensitive cells are in many places but without much detail of their specific morphologies. This comment extends to the use of single cell data. The cells in cluster 4 would need a better visualization (besides being assigned to a general “lateral to the intestine” expressing group). The protonephridia of Figure 4h?

5- I find interesting the pattern of GMP expression. However, I find a bit premature to affirm that (last sentence of “GMP expression in gonadal cells”): “our results indicate that the GMP complement expression is restricted to the germ line in *Stenostomum*”. To me the evidence is just circumstantial. We have no probe of that. Moreover, the authors could have done some (additional) sections and show the morphology of those gonadal-related cells.

All in all, I find the paper excellent. However, I would like to see my questions answered before endorsing it.

Reviewer #2

(Remarks to the Author)

The authors present a detailed molecular and functional description of division-competent cells in the Catenulid *Stenostomum brevipharyngium*, a basal flatworm capable of whole-body regeneration. This new laboratory strain is situated in a phylogenomically informative position, and aspects of its life cycle strategy deviate from expectations: it lacks a

canonical adult pluripotent stem cells (aPSC) population and instead may rely on adult tissue stem cells for tissue homeostasis and the production of new tissues during paratomy (asexual reproduction) and regeneration. This novel finding brings into question whether aPSCs evolved multiple times in evolution –perhaps by convergently co-opted use of “germline multipotency genes” for sustained pluripotency – or whether Catenulids lost aPSCs during evolution. This manuscript cannot answer this question, but of course that doesn't detract from the novelty of this new research organism and its life cycle strategy, nor would the answer detract from the importance of understanding aPSC/neoblast biology in animals that rely on them to produce new tissues. This work lays a solid foundation for interrogating the basis of this complex adult stem cell system, where they will be able to address questions like whether dividing cells are uni- or multipotent? Integrated into the tissue or migratory (perhaps emanating from the lateral tracts adjacent to the gut)? Is cell division coupled to turnover or apoptosis? How are stem cells differentially regulated during homeostasis versus regeneration? And how are germ cells specified, and how do these animals switch between asexual and sexually reproducing states?

The manuscript is well-written and uses precise and accurate language to describe previous findings in the field and their own data, without over-reaching statements. I appreciate the care, intention, and rigor on display throughout this work. Kudos to all the authors for this thought-provoking work. It's a delightful read, and one that is appropriate in quality and completeness for publication in Nature Communications.

My suggestions are minor and for clarity:

Irradiation dosing: Were the dose regimens administered in quick succession, or were there recovery periods between doses (e.g., 100 + 100 + 50 Gy)? This can be clarified in the Materials and Methods text. It appears that 24 hour recovery periods were included (Figure S1), but it should be made clear if this was standard practice for all assays with this notation.

Single nuclei data analysis:

Criteria and data used (e.g. log2FC, padj values, etc) for selecting differentially expressed genes (Supplementary Tables 4 and 5) would be nice to see alongside the homology information and manual data curation.

Figure 4: These HCR+EdU costains are striking. Is quantification of triple positive cells achievable using existing images? The fraction of triple positive cells relative to total EdU cells, along with number of animals screened, would bolster the conclusion, since you are showing rare events with these short pulses. The authors may also consider using a color blind-friendly pseudocoloring scheme (magenta/green vs red/green) to make their figures accessible.

Please ensure the raw bulk and single nuclei sequencing data is deposited and made available to the community at the time of publication.

Reviewer #3

(Remarks to the Author)

The manuscript by Gąsiorowski et al presents exciting and highly valuable new data on the stem cell system of *Stenostomum*, a basally branching flatworm. Using diverse and complementary approaches (EdU assays, irradiation, single cell, HCR, functional studies), the authors very convincingly demonstrate extensive differences to the neoblast based system of planarians and other platyhelminths. They identify a cell cluster (cluster 4) in *Stenostomum* that is proliferative and is transcriptionally closest cell cluster to the planarian neoblasts but remarkably this cluster does not express germline multipotency genes as it does in other platyhelminths. Furthermore, inhibition of GMP genes in *Stenostomum* remarkably does not affect regeneration or asexual reproduction. This is a study we have long been waiting for that helps to put the unusual neoblast system of planarians into evolutionary context. It has broad implications for understanding both the evolution of the planarian neoblast system and the evolution of stem cell systems in regeneration across animals. The data are incredibly exciting, the manuscript is clearly written, and the context and importance are made clear without any overinterpretation of the data. I am very impressed by this study and this manuscript. I have only a few small comments.

Fig 1 e, f: Some of the green arrowheads are hard to distinguish when overlaid on nuclear stain. Consider slight shift in location of the arrowheads or possibly adding a very fine white or dark green outline to the arrowheads that are hard to see.

Irradiation assays methods: Should specify the frequency of checks for these studies. It's not clear what “regular observations” means. Give some sense of this such as “every 1-4 days”, or even “typically every 2-3 days”.

Fig 3 needs to indicate the actual genes that were used for the HCRs. The gene references should be given in the figure legend or there should be a reference in the legend to a supplementary document that gives these gene names. Given the importance of cluster 4, the gene used for the cluster 4 HCR in this figure should be given in the legend, even if other genes are referred to only in a supplementary document that is cited.

Page 4: Regarding the cluster 4 characterization, the authors state that it “lacked GMP components except for a single argonaute paralog”. This is a key finding and the text should make it clear what genes it lacks and/or there should be a reference to Fig 3f where these genes are indicated.

Fig 4 panel n second image. 5 pha should be 5 hpa

Figure 6. This is an excellent summary figure!

Version 1:

Reviewer comments:

Reviewer #1

(Remarks to the Author)

My specific concerns have been all addressed satisfactorily. Moreover, the paper, excellent before, reads even better now. I appreciate the effort of the authors adding new experimental data/figures to the first draft. Thus, and at this point, I endorse the publication of the paper and congratulate the authors for this pioneering work.

Reviewer #2

(Remarks to the Author)

The revised manuscript satisfactorily addresses the reviewers' comments and is now suitable for acceptance and publication in Nature Communications. Thank you for the opportunity to inform the presentation of this impressive body of work.

Reviewer #3

(Remarks to the Author)

The authors have satisfactorily addressed all of my comments, and their edits in response to the other reviewers' comments also seem clearly addressed. My congratulations to the authors on an excellent study.

We would like to thank our reviewers for their critical reading of the manuscript, their constructive suggestions, and explicit expressions of appreciation for our work. As detailed in the following point-by-point responses, we believe that we have addressed all major issues and we believe that the revised manuscript is now ready for publication.

Reviewer #1:

Comment: „*The paper by Gasiowski and collaborators present a challenge to the paradigm that Platyhelminthes regenerate body parts by mobilizing stem-cells (neoblasts). A thorough analysis of the regeneration mechanisms in the catenulid Stenostomum challenge previous assumptions on the presence/use of the neoblasts in the process. In this context, the findings offer important hints into the developmental and evolutionary roles of stem cells. The experimental strategy is well conceived, the findings well described, and the writing clear and focused.*

Here I suggest some issues that (to me) need some clarification; some of them minor corrections. I find a bit complicated to describe them in the absence of line numbers in the draft. They are listed in the order in which they appear in the text:!“

Answer: Thank you for your positive feedback on our manuscript.

Comment: „*1- In the introduction, the authors mentioned that neoblasts “revealed a remarkable degree of conservation between those groups”. I guess the author refer to “similarities”, instead of “Conservation”. Otherwise, the sentence turns to be a bit vague. Conservation of what? (morphology? gene patterns? both??)*“

Answer: Done. The revised text now reads “Originally defined based on morphology [2], molecular and functional characterizations of neoblasts in planarians [5-7], macrostomids [8, 9], and neodermatans (parasitic flatworms) [10, 11], revealed remarkable similarities between those groups.” (lines 36-39)

Comment: „*2- I find that the irradiation experiments, common in other flatworms, need a bit of explanation. For instance, how do we know that irradiation (mostly at high doses) is not randomly killing (or altering) all types of mature cells?*“

Answer: Thank you – we agree that more background is needed. While irradiation causes double-strand DNA breaks irrespective of the cell type, cells in the process of division are particularly sensitive to DNA damage due to the activity of several cell cycle checkpoints. Since the neoblasts are the only dividing somatic cell type in many flatworms, irradiation is the go-to approach for neoblast elimination in the field and multiple studies support its specificity (e.g., irradiation-sensitive genes are largely neoblast-specific). The fact that irradiated *Stenostomum* worms survive around 2-3 weeks even after high doses of irradiation is in line with the assumption that most somatic cells, even if affected by X-rays, can survive and function. Similarly, the lethal phenotype is indicative of the depletion of cell turnover and not the acute effects of irradiation on somatic cell types. Following the reviewer’s comment, we now modified the section “*Stenostomum* tissue dynamics” to better explain this context to

the readers without prior knowledge of the principle of irradiation-induced ablation of the neoblast. The revised text now reads “To assess the contribution of newly generated cells to the regeneration process, we ablated division-competent cells by X-ray irradiation. Irradiation induces double-strand breaks in DNA, which are particularly toxic to dividing cells [41]. Given the privileged division competence of flatworm neoblasts, irradiation has been used extensively for experimental neoblast ablation [6, 9, 42]. Neoblast ablation in macrostomids, planarians and schistosomes blocks regeneration and eventually kills the animals due to the failure of homeostatic cell turnover [9, 11, 42, 43].” (lines 83-89).

Comment: „3- Following my previous question, why (in the section *Irradiation-sensitivity of gene expression*) should we assume that “ribosomal biogenesis may be largely restricted to dividing cells in *Stenostomum*”? Maybe irradiation has specific transcriptional effects on all cells of each individual. In fact, the disrupted morphology of the irradiated specimens’ points to the possibility that many cells are affected/damaged. The conclusion that the transcriptional data is the “first indication of divergent transcriptional signature of dividing cells” is not fully guaranteed (though, still, reasonable).”

Answer: We agree that the depletion of ribosomal genes in transcriptomes of irradiated worms is not definitive evidence for the enrichment of those genes in dividing cells of *Stenostomum*. We therefore changed the wording of the respective text section, which now reads (“...**suggesting** that ribosomal biogenesis **may be** largely restricted...”; lines 120-121). In addition, please note the following manuscript sections fully confirm the initial suggestion—e.g., the expression of ribosomal genes (e.g. FBL) in EdU-positive cells (Figs. 2F, 4B and D) and the expression distribution of ribosomal genes in our single-nuclei atlas (Extended Fig. 4F).

Comment: „4- The fact that the irradiation-sensitive genes map to different regions of the body is illuminating and serves the purpose that they are distributed in different tissues. However, it would have been a nice complement to show a few sections (*semithin*), with a more detailed analysis of the cells involved and their relative positions in the tissues. The double *in situ* with specific tissue markers offers some hints (though, of course, miss the resolution of a good histological section). We are left with the idea that the sensitive cells are in many places but without much detail of their specific morphologies. This comment extends to the use of single cell data. The cells in cluster 4 would need a better visualization (besides being assigned to a general “lateral to the intestine” expressing group). The protonephridia of Figure 4h?”

Answer: Thank you for this suggestion. We performed the suggested semi-thin sectioning of *Stenostomum* following standard protocols and the results of two transverse sections of the stem cell containing trunk region are shown below. Due to the microscopic body size of the worms and the very small size of their cells, the resulting insights into histological details are very limited.

In the top row, we show the raw microscopic images of two representative sections stained with toluidine blue. In the lower row, the same sections are color-labeled showing the areas occupied by gut (magenta), protonephridium (green), and presumptive stem cell clusters (red).

To overcome this limitation, we instead performed triple in situ with molecular markers for stem cells (*C4M1*), the protonephridium (*SLC4A7*), the epidermis (*EM1*), the gut (*GM1*), and germline cells (*piwiA*) in different combinations and imaged them with high Z resolution using confocal microscopy. This approach yields virtual cross-sections through the trunk, with different tissues labeled with respective markers. Although those cross-sections lack information on the ultrastructure of particular cell types, they accurately visualize the position and orientation of stem and germline cells in relation to the other major cell types. These data have been included as Supplementary Figure 6 and now provide the requested detail at least on the tissue distribution of stem- and germ cells.

Comment: „5- I find interesting the pattern of GMP expression. However, I find a bit premature to affirm that (last sentence of “GMP expression in gonadal cells”): “our results indicate that the GMP complement expression is restricted to the germ line in *Stenostomum*”. To me the evidence is just circumstantial. We have no probe of that. Moreover, the authors could have done some (additional) sections and show the morphology of those gonadal-related cells.

Answer: As shown above, at least the standard semi-thin sectioning protocols are of limited utility in *Stenostomum*. The optical cross-sections of *piwiA* stained worms that we added as Supplementary figure 6 now add anatomical context of *piwiA*⁺ cells and we refer to the additional evidence in the text section on GMP genes. In addition, we modified the concluding sentence, which now reads “our results **suggest** that the GMP complement expression is restricted to the germ line in *Stenostomum*.” (lines 299-300). We would further like to point out that we believe that the co-expression of three important GMP components (*piwiA*, *vasa* and *nanos*) and the other lines of evidence that we provide for the germ line restriction of GMP together go beyond "circumstantial" and we hope that we have convinced the reviewer accordingly.

All in all, I find the paper excellent. However, I would like to see my questions answered before endorsing it.“

Thank you. We hope that the revised manuscript now earns your endorsement.

Reviewer #2:

Comment: „The authors present a detailed molecular and functional description of division-competent cells in the Catenulid *Stenostomum brevipharyngium*, a basal flatworm capable of whole-body regeneration. This new laboratory strain is situated in a phylogenomically informative position, and aspects of its life cycle strategy deviate from expectations: it lacks a canonical adult pluripotent stem cells (aPSC) population and instead may rely on adult tissue stems cells for tissue homeostasis and the production of new tissues during paratomy (asexual reproduction) and regeneration. This novel finding brings into question whether aPSCs evolved multiple times in evolution –perhaps by convergently co-opted use of “germline multipotency genes” for sustained pluripotency – or whether Catenulids lost aPSCs during evolution. This manuscript cannot answer this question, but of course that doesn’t detract from the novelty of this new research organism and its life cycle strategy, nor would the answer detract from the importance of understanding aPSC/neoblast biology in animals that rely on them to produce new tissues. This work lays a solid foundation for interrogating the basis of this complex adult stem cell system, where the will be able to address questions like whether dividing cells are uni- or multipotent? Integrated into the tissue or migratory (perhaps emanating from the lateral tracts adjacent to the gut)? Is cell division coupled to turnover or apoptosis? How are is stem cells differentially regulated during homeostasis versus regeneration? And how are germ cells specified, and how do these animals switch between asexual and sexually reproducing states?

The manuscript is well-written and uses precise and accurate language to describe previous findings in the field and their own data, without over-reaching statements. I appreciate the care, intention, and rigor on display throughout this work. Kudos to all the authors for this thought-provoking work. It’s a delightful read, and one that is appropriate in quality and completeness for publication in *Nature Communications*.

My suggestions are minor and for clarity:“

Answer: Thank you – much appreciated!

Comment: *Irradiation dosing: Were the dose regiments administered in quick succession, or were there recovery periods between doses (e.g., 100 + 100 + 50 Gy)? This can be clarified in the Materials and Methods text. It appears that 24 hour recovery periods were included (Figure S1), but it should be made clear if this was standard practice for all assays with this notation.*“

Answer: Done. Following the reviewer’s suggestion, we now explicitly state the recovery periods between doses in the materials and methods section (lines 616-617).

Comment: „Single nuclei data analysis: Criteria and data used (e.g. log2FC, padj values, etc) for selecting differentially expressed genes (Supplementary Tables 4 and 5) would be nice to see alongside the homology information and manual data curation.“

Answer: Done. We added log2FC columns showing the relative enrichment of each of our manually selected markers in particular clusters and subclusters to supplementary tables 4 and 5. Originally, we chose the genes used for annotation manually by examining top markers for particular clusters with random forest classifiers on each Leiden cluster, as stated in the material and methods section.

Comment: „Figure 4: These HCR+EdU costains are striking. Is quantification of triple positive cells achievable using existing images? The fraction of triple positive cells relative to total EdU cells, along with number of animals screened, would bolster the conclusion, since you are showing rare events with these short pulses. The authors may also consider using a color blind-friendly pseudocoloring scheme (magenta/green vs red/green) to make their figures accessible.“

Answer: Done and thank you for the suggestion. We quantified triple and double positive cells and the results of this quantification are now shown in Supplementary Figure 6 and referred to in the manuscript. As for color blind-friendly pseudocoloring schemes, we struggled with their inherent limitations for visualizing triple color in situ and therefore chose to stick with the RGB data visualization. However, we would like to point out that we are deliberately showing each color channel as a separate sub-panel, which we hope will allow visually impaired readers to judge the spatial overlap between channels.

Comment: „Please ensure the raw bulk and single nuclei sequencing data is deposited and made available to the community at the time of publication.“

Done. All the raw data has been deposited in the publicly available repositories (irradiation transcriptomes: NCBI, BioProject PRJNA1149834 will be released on the 15th of September; single-cell transcriptomic dataset: PRJNA1156255). All the reference numbers to the deposited datasets are now provided in the manuscript (lines 816-823).

Reviewer #3:

Comment: „The manuscript by Gqsiorowski et al presents exciting and highly valuable new data on the stem cell system of *Stenostomum*, a basally branching flatworm. Using diverse and complementary approaches (EdU assays, irradiation, single cell, HCR, functional studies), the authors very convincingly demonstrate extensive differences to the neoblast based system of planarians and other platyhelminths. They identify a cell cluster (cluster 4) in *Stenostomum* that is proliferative and is transcriptionally closest cell cluster to the planarian neoblasts but remarkably this cluster does not express germline multipotency genes as it does in other platyhelminths. Furthermore, inhibition of GMP genes in *Stenostomum* remarkably does not affect regeneration or asexual reproduction. This is a study we have long been waiting for that helps to put the unusual neoblast system of planarians into evolutionary context. It has broad implications for understanding both the evolution of the planarian neoblast system and the evolution of stem cell systems in regeneration across animals. The data are incredibly exciting, the manuscript is clearly written, and the context and importance are made clear without any overinterpretation of the data. I am very impressed by this study and this manuscript. I have only a few small comments.“

Answer: Thank you very much for your explicit support – much appreciated!

Comment: “Fig 1 e, f: Some of the green arrowheads are hard to distinguish when overlaid on nuclear stain. Consider slight shift in location of the arrowheads or possibly adding a very fine white or dark green outline to the arrowheads that are hard to see.“

Answer: Done. We improved the visibility of the arrowheads in Fig. 1 e and f.

Comment: “Irradiation assays methods: Should specify the frequency of checks for these studies. It’s not clear what “regular observations” means. Give some sense of this such as “every 1-4 days”, or even “typically every 2-3 days”.“

Answer: Done and thank you for the suggestion. We now explicitly specify the frequency of observations on which the survival curves are based in the respective materials and methods section (lines 619-621). Further details on observation timing are also available in Supplementary Table 7.

Comment: “Fig 3 needs to indicate the actual genes that were used for the HCRs. The gene references should be given in the figure legend or there should be a reference in the legend to a supplementary document that gives these gene names. Given the importance of cluster 4, the gene used for the cluster 4 HCR in this figure should be given in the legend, even if other genes are referred to only in a supplementary document that is cited.“

Answer: Done and thank you for catching this. We now added information on the HCR markers in Fig. 3 to the legend and Supplementary Table 3.

Comment: “Page 4: Regarding the cluster 4 characterization, the authors state that it “lacked GMP components except for a single argonaute paralog”. This is a key finding and the text

should make it clear what genes it lacks and/or there should be a reference to Fig 3f where these genes are indicated.“

Answer: The particular sentence that the reviewer refers to here regards our SAMap comparison of cluster 4 with the neoblast of *S. mediterranea*. To clarify this, we added a reference to the supplementary table and the gene ID of the aforementioned AGO paralog to specify which gene we refer to. Please note that Fig 3F is already referred to in the text when it comes to the analysis of the expression of GMP components of *Stenostomum* in particular cell types.

Comment: *“Fig 4 panel n second image. 5 pha should be 5 hpa”*

Answer: Done.

Comment: *“Figure 6. This is an excellent summary figure!”*

Answer: Thank you!